# *Euglena gracilis* Enhances Innate and Adaptive Immunity through Specific Expression of Dectin-1 in CP-Induced Immunosuppressed Mice

**DOI:** 10.3390/nu16183158

**Published:** 2024-09-18

**Authors:** Hwan Hee Lee, Ji-Yeon Seong, Hyojeung Kang, Hyosun Cho

**Affiliations:** 1College of Pharmacy, Duksung Women’s University, Seoul 01369, Republic of Korea; oeo3oeo@gmail.com; 2Duksung Innovative Drug Center, Duksung Women’s University, Seoul 01369, Republic of Korea; sjyne@duksung.ac.kr; 3Vessel-Organ Interaction Research Center, VOICE (MRC), Cancer Research Institute, College of Pharmacy, Kyungpook National University, Daegu 41566, Republic of Korea

**Keywords:** *E. gracilis*, β-1,3-glucan, immune-enhancing effect, immunosuppressed mice, dectin-1, cyclophosphamide

## Abstract

Background: *Euglena gracilis* (*E. gracilis*), a species of unicellular algae, can accumulate large amounts of β-1,3-glucan paramylon, a polysaccharide, in its cytoplasm and has recently attracted interest as a bioproduct due to its various health benefits. In this study, the immune-enhancing effect of *E. gracilis* powder (EP) was investigated in vitro and in vivo. Methods: In vitro, the production of NO and cytokines and the mechanism of the signaling pathway of β-1,3-glucan were identified in RAW264.7 cells. In vivo, cyclophosphamide-induced (CP-induced) immunosuppressed C57BL/6 female mice were orally administered with three different concentrations (100, 300, and 600 mg/kg) of EP daily. After 14 days, the organs and whole blood were collected from each animal for further study. Results: The weight loss of CP-treated mice was reversed by treatment with EP to levels comparable to those of control mice. In addition, the frequencies of NK1.1^+^, CD3^+^, CD4^+^, CD8^+^, and B220^+^ in immune cells isolated from the spleen were increased by EP treatment compared with water or RG. The secretion of TNF-α, IFN-γ, and IL-12 from splenocytes was also increased by EP treatment, as was the level of IgM in the serum of the mice. Finally, EP treatment specifically upregulated the expression of dectin-1 in the liver of CP-treated mice. Conclusions: *E. gracilis* could be a good candidate for a natural immune stimulator in the innate and adaptive response by secreting TNF-α, IFN-γ, and IL-12 through stimulating dectin-1 expression on the surface of immune cells.

## 1. Introduction

The immune system consists of innate and adaptive immunity, and it plays a critical role in resisting pathogenic infection and in maintaining homeostasis. Innate immunity is the first line of defense against pathogens and includes phagocytes, which consist of neutrophils, dendritic cells, blood monocytes, and tissue macrophages. Phagocytes identify pathogens through pattern recognition receptors (PRRs), including Toll-like receptors (TLRs). This recognition can lead to the direct killing of pathogens or facilitate the activation of T cells, which are integral to the adaptive immune response, by presenting antigens [1]. Lymphocytes include natural killer (NK) cells, which are part of the innate immune system, and T cells and B cells, which are responsible for adaptive immune responses. NK cells are most abundant in the peripheral blood and are also present in other organs [2]. Their function is to directly kill virus-infected or transformed cells without prior sensitization [3], and they have recently been shown to produce several cytokines that help other immune cells [4]. In adaptive immunity, T or B lymphocytes can remember previous invaders and develop an intense response to a second invasion. Therefore, the coordination between different immune cells is very important to elicit an appropriate immune response to a foreign agent [5,6].

*E. gracilis* is a species of unicellular algae found in most freshwater biotopes and is characterized by being mixotrophic, capable of both prototrophic and heterotropic feeding [7,8]. Recent studies have highlighted the potential of bioproducts derived from *E. gracilis* due to its diverse range of components, including the main component β-1,3-glucan paramylon, as well as amino acids, provitamins, and lipids [9,10,11]. In particular, β-1,3-glucan paramylon is a reserve polysaccharide and can accumulate in large amounts (>80%) depending on the cultivation conditions of *E. gracilis* [8,12]. Other derived β-glucans (e.g., yeast, oat) are already known to have various bioactivities, including immunostimulatory and antimicrobial effects in animals and humans [13,14,15].

Dectin-1 is a C-type lectin receptor (CLR) that recognizes β-glucans from fungi, plants, and algae [16]. It is expressed mainly on myeloid cells but can also be found on some lymphocytes [17]. Binding of β-glucans to dectin-1 on myeloid cells results in the production of several cytokines and reactive oxygen species (ROS) involved in myeloid cell activation reactions [18,19]; however, this depends on the structural conformations of the β-glucans.

Cyclophosphamide (CP) is a drug used in chemotherapy to treat many cancers, including lymphoma, multiple myeloma, and breast cancer [20]. Its mechanism in cancer is to interfere with protein synthesis through DNA and RNA cross-linking, resulting in apoptosis of cancer cells and inhibition of cell cycle [21,22]. However, CP can reduce the number of lymphocytes such as T cells and B cells [21,22]. In this respect, it is often used as a potent immunosuppressive agent in animal experiments to investigate the immunomodulatory effects of certain substances in vivo [23,24].

Several studies have demonstrated the effect of *E. gracilis* on innate and adaptive immunity through in vitro and in vivo experiments, but the mechanism is still unclear. The immune system is critical to maintaining body homeostasis, so it must be controlled to ensure that immune responses do not become excessively elevated or depressed. CP is known to induce the depression of lymphocytes in mice, which disrupts the homeostasis of the immune system. In light of this, we investigated the immune-enhancing effect of EP containing large amounts of β-1,3-glucan paramylon using CP-induced immunosuppressed mice.

## 2. Materials and Methods

### 2.1. Specimen Preparation

β-1,3-glucan derived from *E. gracilis* was sourced from Sigma-Aldrich Inc. (St. Louis, MO, USA). For in vitro study, β-1,3-glucan was treated with 0.5 mol/L NaOH and stirred until a homogeneous solution was achieved. It was then precipitated using cold 98% ethanol and collected by centrifugation at 12,000× *g* for 10 min at 4 °C. The specimen was suspended in 40 mL of deionized water and subsequently diluted, following the method described in [13]. BioGlena^TM^, EP containing over 50% β-1,3-glucan, was acquired from JUYEONG NS Co., Ltd. (Seoul, Republic of Korea), which is supplied by Algatechnologies Ltd. (Kibbutz Ketura, Israel). For the in vivo study, EP was dissolved in drinking water.

### 2.2. Cell Culture and Treatment

The mouse macrophage cell line RAW264.7 was obtained from the Korea Cell Line Bank (KCLB, Seoul, Republic of Korea). The cells were maintained in Dulbecco’s modified Eagle’s medium (DMEM, Gibco, Grand Island, NY, USA) supplemented with 10% fetal bovine serum (FBS, Gibco) and 1% penicillin and streptomycin (Gibco) at 37 °C in a humidified environment with 5% CO_2_. For in vitro experiments, the cells were seeded into plates and incubated overnight (18–24 h) before being treated with β-1,3-glucan for 24 h. The supernatant was then collected for cytokine analysis, while the cell lysates were prepared for studying the underlying mechanism.

### 2.3. CCK-8 Assay

To assess the cytotoxicity of β-1,3-glucan on RAW264.7 cells, the cells were seeded into a 96-well flat-bottom plate and exposed to lipopolysaccharide (LPS) at 1 μg/mL or varying concentration of β-1,3-glucan (0, 50, 100, 250, 500, and 1000 μg/mL) for 24 h. Following this, the CCK-8 solution was added to each well, and the cells were incubated for an additional 3 h. The absorbance was then read at 490 nm using a microplate reader (BMG Labtech, Ortenberg, Germany).

### 2.4. Nitric Oxide (NO) Assay

Cell-free supernatants were collected from RAW264.7 cells that had been treated with LPS or varying concentrations of β-1,3-glucan (0, 50, 100, 250, and 500 μg/mL) for 24 h to assess the production of nitric oxide (NO) metabolites. The nitrite/nitrate assay kit (Sigma-Aldrich) was used following the protocol provided by the manufacturer. The absorbance was measured at 540 nm using a microplate reader.

### 2.5. Cytokine Analysis

Supernatants from RAW264.7 cells treated with LPS or various concentrations of β-1,3-glucan, as well as from mouse splenocytes incubated with or without concanvalin A (ConA, 10 μg/mL) for 72 h, were collected for cytokine analysis. Levels of TNF-α and IL-1β were measured using a mouse Uncoated ELISA kit (Invitrogen, Waltham, MA, USA), while IL-6, IL-12, and IFN-γ were quantified using a mouse ELISA kit (BD Biosciences, Franklin Lakes, NJ, USA). All procedures followed the protocols provided by the manufacturers. Absorbance readings were taken at 450 nm using a microplate reader (BMG Labtech).

### 2.6. Western Blot Analysis

RAW264.7 cells were exposed to LPS or different concentrations of β-1,3-glucan (50, 100, 250, and 500 μg/mL) for 24 h. After treatment, the cells were lysed using a protein extraction buffer (Intron, Seoul, Republic of Korea) containing phosphatase inhibitors. Protein concentration was determined using the Bradford (Coomassie blue) assay. Proteins were then separated by electrophoresis, transferred onto polyvinylidene fluoride (PVFD) microporous membrane with a 0.2 μm pore (Millipore, Burlington, MA, USA), and subjected to immunoblotting with primary and secondary antibodies. Protein bands were visualized using the Chemi-Doc imaging system (Millipore) and enhanced chemiluminescence detection solution. The primary antibodies used included anti-TLR6, anti-pERK (p42/p44), anti-pNF-κB (all from Cell Signaling, Danvers, MA, USA), anti-dectin-1 (Abcam, Cambridge, UK), and anti-β-actin (Sigma-Aldrich, St. Louis, MO, USA). The secondary antibodies, anti-mouse IgG and anti-rabbit IgG, were obtained from Cell Signaling Technology (CST) Inc. (Danvers, MA, USA).

### 2.7. In Vivo Experiments Using Immunosuppressed C57BL/6 Mice

All in vivo studies were conducted following the guidelines set by the National Research Council’s Guide for Care and Use of Laboratory Animals. The study protocol received approval from the Animal Experiments Committee of Duksung Women’s University (permit number: 2024-004-026). Female C57BL/6 mice, 5 weeks old, were obtained from JABIO Inc. (Seoul, Republic of Korea). Prior to the experiment, all mice were acclimated in the laboratory for one week under controlled conditions, including a temperature of 23 ± 2 °C and a 12-h light/dark cycle. The mice were randomly divided into six groups (n = 8 per group) and received intraperitoneal injections of 150 mg/kg of CP on days 1 and 3, except for the control group (NC) which received only drinking water. The groups were as follows: (1) control with drinking water (NC), (2) water with CP, (3) 10 mg/kg red ginseng (RG) with CP, (4) 100 mg/kg EP with CP, (5) 300 mg/kg EP with CP, and (6) 600 mg/kg EP with CP. Each test substance was administered orally once daily for 14 days. Throughout the study, mice were weighed daily, and at the end of the experiment, the weight of the liver, spleen, and thymus were recorded.

### 2.8. Complete Blood Cell Count (CBC) Analysis

After 14 days, the mice were euthanized, and whole blood samples were collected for complete blood count (CBC) analysis. The total white blood cell (WBC) count, as well as the percentages of neutrophil (NEU), lymphocyte (LYM), monocyte (MONO), eosinophil (EOS), and basophil (BASO), were measured. Additional parameters such as red blood cell (RBC) count, platelet count, hemoglobin concentration (HGB), mean corpuscular volume (MCV), mean corpuscular hemoglobin (MCH), and mean corpuscular hemoglobin concentration (MCHC) were also assessed. These measurements were performed using an automated hematology analyzer (XN-100, Sysmex, Kobe, Japan).

### 2.9. Antibodies (IgM, IgG) Analysis

For antibody analysis, serum was separated from the whole blood of mice, and splenocytes were isolated from their spleen. The levels of IgM and IgG antibodies were measured using a mouse Uncoated ELISA (Invitrogen). The absorbance was recorded at 450 nm using a microplate reader (BMG Labtech).

### 2.10. Splenocyte Proliferation Assay

Splenocyte proliferation in response to EP was assessed using Cell Counting Kit-8 (CCK-8, Dojindo, Japan). Briefly, splenocytes were seeded in a 96-well flat-bottom plate at a density of 1 × 10^5^ cells per well, with or without concanavalin (ConA, 10 μg/mL), and incubated for 72 h. After incubation, CCK-8 solution was added to each well and allowed to incubate for another 3 h. The absorbance was measured at 450 nm using a microplate reader.

### 2.11. Cell Surface Antigens Analysis by Flow Cytometry

Spleen cells from mice were stained with a combination of mouse anti-NK1.1-APC and anti-CD3-PE, a combination of anti-CD4-PE and anti-CD8-APC, and anti-B220-PE antibodies (all from BD Biosciences, Franklin Lakes, NJ, USA) for 30 min in the dark. After staining, the cells were analyzed using flow cytometry (Novocyte Flow Cytometer, ACEA Biosciences, Santa Clara, CA, USA). The positive cell populations were identified by comparing the results with predefined cutoff values obtained from unstained control cells, as previously described [23].

### 2.12. Immunohistochemistry (IHC) Staining

Mouse liver tissues were fixed using the paraffin embedding technique. Prior to antibody staining and blotting, the paraffin was removed with xylene, followed by dehydration with ethanol and treatment with 3% hydrogen peroxide (H_2_O_2_) diluted in methanol. Subsequently, the tissues were incubated with the primary antibody (m-dectin-1), followed by the secondary antibody. The tissues were then treated with avidin/biotin and a DAB substrate. After staining with hematoxylin, the tissues were mounted. Protein expression in the tissues was examined using light microscopy at magnifications of 40× and 200×.

### 2.13. Statistics

To ensure the significance of results in vitro experiments, all assays were conducted in triplicate, and the data are presented as mean ± SD. Both in vitro and in vivo data were analyzed using GraphPad Prism version 5.03 for Windows (GraphPad Software, www.graphpad.com (accessed on 15 September 2024)). Mean comparisons were performed using *t*-tests or one-way ANOVA followed by Tukey’s multiple comparison test. A *p*-value of less than 0.05 was considered statistically significant.

## 3. Results

### 3.1. Production of Nitric Oxide (NO) in RAW264.7 Cells by β-1,3-Glucan

The production of NO metabolites by macrophage in response to β-1,3-glucan was investigated in RAW264.7 cells using a nitrite/nitrate assay kit. We first identified the cell cytotoxicity to β-1,3-glucan using the CCK-8 assay. β-1,3-glucan in RAW264.7 cells was not cytotoxic at any treated concentrations (Figure 1A). The NO production in response to β-1,3-glucan in RAW264.7 was then measured. When cells were treated with different concentrations (50, 100, 250, and 500 μg/mL) of β-1,3-glucan for 24 h, NO production was significant at 250 and 500 μg/mL of β-1,3-glucan compared to the untreated group (untreated 13.90 μM, 250 μg/mL 18.98 μM, 500 μg/mL 19.16 μM), as shown in Figure 1B. 

### 3.2. Specific Production of Pro-Inflammatory Cytokine TNF-α via a Stimulation of TLR-6 and Dectin-1 Proteins on a Surface of RAW264.7 Cells by β-1,3-Glucan

The production of inflammatory cytokines in macrophages is a hallmark of the innate immune response. Therefore, we investigated the cytokine production triggered by β-1,3-glucan in RAW264.7 cells. For cytokine analysis, the cell-free supernatants were collected from cells treated with β-1,3-glucan for 24 h. Interestingly, β-1,3-glucan significantly stimulated the production of TNF-α (Figure 2A), but not that of IL-6 and IL-1β (Appendix A). The presence of β-1,3-glucan significantly increased the expression of TLR6 and dectin-1 on the cell surface (Figure 2B,C), in addition to increasing the expression of pERK and pNF-κB, although not in a concentration-dependent manner. 

### 3.3. Effect of EP on the Weight of Body and Organs in the Immunosuppressed Mice

To determine the immune-enhancing effect of *E. gracilis*, which contains high amounts of β-1,3-glucan in vivo, the immunosuppressed mice induced by intraperitoneal injection of CP (100 mg/kg) were used. Mice received CP on day 1 and day 4 from the first day of oral administration of EP. CP induced a significant decrease in the body weight of mice on day 6 compared to the water-only group (NC) (Figure 3A). By day 8, the mice were in a recovery of body weight except for those in the CP group, and in addition, all EP groups showed a significant difference compared to the CP group (Figure 3B). All the animals were sacrificed after 14 days of oral administration. The liver weight of the mice in the CP group was significantly increased compared with that of the mice in the NC and EP groups, and the spleen weight was also increased (Figure 3D,E). However, the thymus weight did not differ among NC, CP, and RG groups, but was significantly different in EP groups (Figure 3F). 

### 3.4. CBC Analysis of Whole Blood from the Mice Treated with EP

The total white blood cell (WBC) count was lower in all groups treated with CP than in the untreated group (NC), as shown in Figure 4A and Table 1. However, the frequency of myeloid cells (neutrophil; NEU, monocyte; MONO, eosinophil; EOS, basophil; BASO) was relatively increased in the CP-treated groups compared to the untreated group, with the neutrophil level being especially significant (Figure 4B, NEU; NC 14.95%, CP 41.40%, RG 10 mg/kg 37.29%, EP 100 mg/kg 31.99%, EP 300 mg/kg 33.10%, EP 600 mg/kg 36.53%). While the frequency of lymphoid cells (LYM) was significantly reduced in the CP-treated groups compared to the untreated group (NC), it was reduced to a significantly lesser extent in the RG and EP groups (Figure 4C; NC 75.90%, CP 37.20%, RG 10 mg/kg 53.55%, EP 100 mg/kg 56.75%, EP 300 mg/kg 54.00%, EP 600 mg/kg 53.23%). The red blood cell (RBC) count and RDW (%) level of mice in the CP group were significantly different from those in the NC group, but not in the other groups (Figure 4D,E). The platelet (PLT) count was significantly increased in CP-treated mice compared to untreated mice (Figure 4F). 

### 3.5. Increased Frequency of Lymphoid Cells in Mouse Spleen by EP

The total frequency of lymphocytes in the CBC of CP-treated mice was significantly decreased, but the proliferation of cells isolated from the spleen of mice was not different among all groups (Figure 5A). The frequencies of NK1.1^+^, CD3^+^, CD4^+^, and CD8^+^ cells were higher in all EP groups than in the others, as shown in Figure 5B–E (NK1.1^+^: NC 3.3%, CP 6.1%, RG (10 mg/kg) 8.2%, EP (100 mg/kg) 8.9%, EP (300 mg/kg) 9.5%, EP (600 mg/kg) 9.2%; CD3^+^: NC 32.9%, CP 29.5%, RG (10 mg/kg) 35.5%, EP (100 mg/kg) 35.3%, EP (300 mg/kg) 37.6%, EP (600 mg/kg) 37.9%; CD4^+^: NC 18.9%, CP 16.4%, RG (10 mg/kg) 21.5%, EP (100 mg/kg) 24.3%, EP (300 mg/kg) 24.9%, EP (600 mg/kg) 27.0%; CD8^+^: NC 11.6%, CP 10.9%, RG (10 mg/kg) 12.0%, EP (100 mg/kg) 13.3%, EP (300 mg/kg) 15.9%, EP (600 mg/kg) 15.2%). The frequency of B220^+^ cells was decreased by CP treatment but was significantly restored by RG or EP treatment compared to CP treatment only (Figure 5F; NC 54.3%, CP 21.7%, RG (10 mg/kg) 38.8%, EP (100 mg/kg) 41.4%, EP (300 mg/kg) 41.2%, EP (600 mg/kg) 40.3%). 

### 3.6. Increased Production of Cytokines (TNF-α, IFN-γ, IL-12) and IgM Antibody in Immunosuppressed Mice by EP

The effect of EP on the production of cytokines and antibodies was analyzed in splenocytes or serum of mice by an ELISA assay. The splenocytes isolated from the mouse spleen were treated with or without ConA for 72 h, and then the cell-free supernatants were collected. Cytokines were produced only in the presence of ConA. TNF-α production was significantly increased by treatment with RG or EP compared to the NC group (Figure 6A). The production of IFN-γ and IL-12 cytokines was reduced in CP-treated mice, but it was significantly increased by RG or EP treatment (Figure 6B,C). In addition, the production of IgM antibody in splenocytes of CP-treated mice was low compared to untreated mice with or without ConA, and it was significantly increased by RG or EP treatment compared to CP-treated group (Figure 6D). In the serum, IgM levels were significantly lower with CP treatment but were significantly increased by EP treatment (Figure 6E).

### 3.7. Increased Expression of Dectin-1 in Liver Tissue of Immunosuppressed Mice by EP

In an in vitro study, β-1,3-glucan induced a significant increase in dectin-1 expression in RAW264.7 cells compared to the control (Figure 2E). Therefore, dectin-1 expression in mouse liver tissue was identified by immunohistochemistry (IHC) staining. The expression of dectin-1 was lower in the mouse liver in the CP group than in the untreated group (NC group). RG treatment didn’t affect the expression of dectin-1, but EP treatment increased it in a concentration-dependent manner (Figure 7). 

## 4. Discussion

In this study, the immune-enhancing effect of EP was investigated using CP-induced immunosuppressed mice. *E. gracilis*, a unicellular alga, can accumulate large amounts of β-1,3-glucan (Paramylon) in its cytoplasm [12], which has recently attracted interest for bioproduct development due to its multiple health benefits [25]. β-glucan is a highly conserved group of glucose polymers and is a structural component of various organisms including fungi, algae, bacteria, and plants. Several studies have reported the various biological functions of β-glucan including immunostimulatory, antioxidant, and antitumor effects [10,13,26]. These effects depend on the structure of β-glucan, such as molecular weight, conformation, and modification [27]. Paramylon is a high molecular weight, linear (unbranched) polysaccharide polymer that has recently been explored for product development in many industries due to its unique structure [28]. The immunostimulatory effect of *E. gracilis*, including its paramylon β-glucan, has been reported in several studies, but the biological mechanisms in animals or humans are not yet well understood.

Reactive oxygen species (ROS) and nitric oxide (NO) are redox molecules that serve as key mediators of innate and adaptive immunity. The production of ROS and NO can trigger specific signals in monocytes and participate in the activation of T cells [29]. In porcine leukocytes, only β-glucan from *E. gracilis* and yeast induced ROS production, while other β-glucan (e.g., Laminarin, Scleroglucan) did not [30]. Our study also showed that β-1,3-glucan extracted from *E. gracilis* was able to generate NO metabolites in the mouse macrophage cell RAW264.7 (Figure 1B). NO is synthesized by many immune cells, including macrophages, and triggers inflammatory signals that lead to the secretion of pro-inflammatory cytokines such as TNF-α, IL-1β, IL-6, and IFN-γ [31,32]. RAW264.7 macrophages treated with *E. gracilis*, of which paramylon is a major component, in a concentration-dependent manner secreted TNF-α and IL-6 [33]. Oral administration of paramylon β-1,3-glucan to mice challenged with the influenza virus H1N1 increased survival and cytokine levels (IFN-γ, IL-1β, IL-6, IL-10, and IL-12) compared to untreated mice [34]. This study found that paramylon β-1,3-glucan induced specific production of only TNF-α but not other cytokines (IL-1β, IL-6, and IL-12, not Supplementary Results) in RAW264.7 cells (Figure 2A). Furthermore, treatment of RAW264.7 cells with paramylon β-1,3-glucan (250 and 500 μg/mL) induced the expression of TLR6 and dectin-1 on the cell surface, whereas LPS (1 μg/mL) significantly increased the expression of TLR2/6 on the cell surface (Figure 2B,C, Appendix A). Several carbohydrate molecules (e.g., mannan, glucans) bind to pathogen recognition receptors (PRRs) on the surface of immune cells and induce proinflammatory cytokine signaling through the NF-κB and the MAP kinase pathways [35]. Glucans induce an immune response by binding to lectin receptors, one of the PRRs, and in particular, β-glucans recognize detin-1 in cooperation with TLR2 on the surface of macrophages [36]. β-glucan signaling is mediated by multiple receptors that ultimately activate transcription factors such as NF-kB, phospholipase C (PLC), and MAPK [37,38,39]. Similarly, the binding of paramylon β-1,3-glucan to its receptors increased the expression of pERK and pNF-κB, although not in a concentration-dependent manner (Figure 2D,E).

Cyclophosphamide (CP) is commonly used as an anticancer chemotherapeutic agent, but high doses of CP can cause acute bone marrow depression [23]. When injected into mice, CP can suppress the function of the thymus and spleen. The thymus and spleen are secondary immune organs and are involved in the development of cellular and humoral immunity. C57BL/6 mice injected with CP had decreased body weight, thymus and spleen sizes, and WBC, RBC, and PLT counts [40]. The effect of *E. gracilis* containing its major component on innate and adaptive immune responses has been reported in several studies. In this study, all mice injected with CP lost weight compared to control mice. Interestingly, mice that were orally administered RG and EP lost less weight on day 8 than those that were not (Figure 3B). RG is commonly used in many studies as a positive sample due to its immune-enhancing effects [41]. After 14 days of oral administration, mice were dissected and the weights of the liver, spleen, and thymus were compared, and the weights of the liver and spleen were increased in the CP group compared with the control group. The liver and spleen weights of mice orally administered RG and EP were not significantly different from the control group (Figure 3E,F). Meanwhile, the thymus weight of mice in the CP group was slightly reduced compared with the control. Previously, several studies reported that CP injection decreased the weight of the spleen and thymus in mice [42]; however, the recent observation showed that the change in the weight of the spleen varied according to the dose of CP, but not that of the thymus [40], which was consistent with our results. Interestingly, the thymus of the mice in all EP groups weighed more than that of the mice in the other groups. The total WBC count was reduced in all CP-treated mice compared to controls (Table 1 and Figure 4A), although the frequency of lymphocytes was somewhat restored in mice orally treated with RG and EP and was significantly different from the CP group (Figure 4C). Lymphocytes play an important role in effective defense against infection and control of disease (e.g., cancer) and are involved in homeostasis, where NK, B, and T cells are present [43]. NK cells, belonging to innate immunity, are important lymphocytes because of their ability to directly kill virus-infected or transformed cells and to help other immune cells by producing cytokines such as IFN-γ [3,4,44]. When NK cells encounter virus-infected cells, they produce granules, such as perforin and granzyme B, to induce apoptosis of virus-infected cells [45]. Mice deficient in perforin or IFN-γ could not control viral infection in the spleen and liver [46]. Therefore, in initiating defense against pathogens, the action of NK cells may be very important in the innate and adaptive immune system. A recent study showed that when *E.gracilis* was supplied to healthy subjects for 8 weeks, it enhanced NK activity compared to a group without supplementation [47]. In addition, they reported that the supplementation of *E. gracilis* increased the serum concentration of IFN-γ, IL-2, IL-10, and IL-12 in participants. In this study, EP significantly increased the frequency of NK1.1^+^ in the spleen of mice compared to other groups of mice (Figure 5B). T lymphocytes, which are involved in cell-mediated immune response, differentiate into the main subtypes, CD4^+^ T cells, CD8^+^ T cells, and T regulatory cells [48]. CD4^+^ T cells are helper cells that secrete cytokines, such as IFN-γ, to help activate other cells. CD8^+^ T cells are called effector cells because they can kill infected or transformed cells. The frequency of T lymphocytes (CD3^+^, CD4^+^, CD8^+^) was slightly reduced in the CP group, and the frequency of B220^+^ cells, which represent B lymphocytes, was significantly reduced (Figure 5C–F). Oral administration of EP resulted in relatively higher frequencies of CD3^+^, CD4^+^, and CD8^+^ T cells than no treatment. In addition, the frequency of B220^+^ cells in the spleen of mice in the RG- and EP-treated groups was significantly increased compared with the CP group. Co-inoculation with β-glucan oligosaccharide increases the population of CD4^+^ and CD8^+^ T cells in lymphoid and non-lymphoid tissues of mice immunized with pB144 DNA vaccine [49]. In vivo, ConA stimulation upregulated the mRNA expression of IFN-γ, IL-10, IL-12Rβ1, IL-1β, and IL-2 in spleen cells of β-glucan- or euglena-treated groups [50]. IFN-γ and IL-2 are the major cytokines associated with T-cell activation, with IFN-γ, IL-2, and IL-12 involved in Th1 cell differentiation and IL-4 and IL-10 involved in Th2 cell differentiation [51]. Th1 cells primarily enhance cell-mediated immunity, whereas Th2 cells enhance humoral immunity. This study found that *E. gracilis* powder promoted the secretion of several inflammatory cytokines, including TNF-α, IL-12, and IFN-γ from splenocytes (Figure 6A–C). B lymphocytes are derived from bone marrow and migrate to the spleen, where they develop into mature B cells. When naïve B cells are activated by an antigen (with the help of T_helper_ cells), they proliferate and differentiate into antibody-secreting effector cells (plasma cells). The primary response to an antigen typically produces more IgM than IgG antibodies, and secondary response relatively increases the level of IgG [52]. CP-treated mice had significantly reduced levels of IgM and IgG produced by spleen cells compared to controls (Figure 6D,F). Dietary supplementation with β-1,3-glucan derived from *E. gracilis* increased the levels of IgG, IgM, and IgA in the serum of the animals [53]. In this study, oral administration of EP to mice rescued IgM levels in serum or secreted from splenocytes, but did not affect IgG levels.

Dectin-1 is a C-type lectin receptor found mainly on myeloid cells such as dendritic cells and macrophages, as well as certain lymphoid cells. It acts as a transmembrane recognition receptor (PRR) that specifically binds to various β-1,3-glucans and polysaccharides [54]. When β-1,3-glucans bind to dectin-1 on the surface of dendritic cells or macrophages, it triggers the production of various inflammatory cytokines, including TNF-α, IL-12, and IFN-γ, which enhance adaptive immune responses [13,55]. As noted earlier, β-1,3-glucan derived from *E. gracilis* triggered the specific expression of dectin-1 in mouse macrophage RAW264.7 (Figure 2C). Dectin-1 is broadly expressed on monocytes across different tissues, especially in the liver and lungs [56]. More recently, studies have shown that dectin-1 mRNA expression was suppressed in the liver and lungs of CP-treated mice and that *E. gracilis* provided protection against CP-induced suppression of dectin-1 expression [57], which was consistent with our results (Figure 7). CP significantly decreased the expression of dectin-1 in liver tissues of mice compared to control; however, oral administration of EP restored the expression of dectin-1 in the mouse liver, but this was not observed in RG-treated mice. This result demonstrated that EP could activate the adaptive immune response by inducing the specific expression of dectin-1.

## 5. Conclusions

This study demonstrated that oral administration of EP to mice helped maintain a high frequency of lymphocytes and upregulate IgM antibody levels through the production of several inflammatory cytokines in CP-induced immunosuppressed mice. In addition, we found that *E. gracilis* induced the specific expression of dectin-1 in mice. Taken together, these results suggest that *E. gracilis* could be used as a potential natural immune stimulator to enhance innate and adaptive immune responses.

## Figures and Tables

**Figure 1 nutrients-16-03158-f001:**
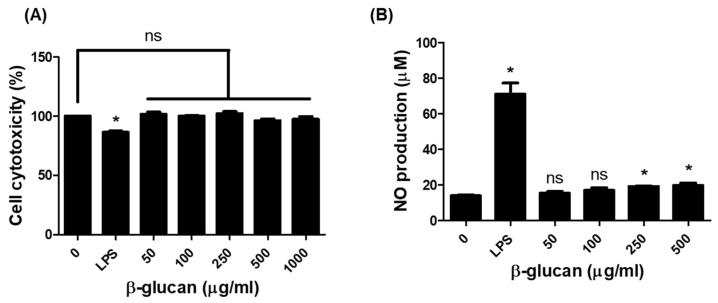
Production of NO metabolites by β-1,3-glucan in RAW264.7 cells. Cells were incubated with a series of concentrations of β-1,3-glucan (0, 50, 100, 250, 500, and 1000 μg/mL) or LPS (1 μg/mL) for 24 h. After the incubation, the (**A**) cytotoxicity by CCK-8 assay and (**B**) NO production in the supernatants collected from the cells by a nitrite/nitrate assay kit were determined. All data are presented as means ± SD, and experiments were performed at least three times. * *p* < 0.05 vs. untreated; ns: not significant.

**Figure 2 nutrients-16-03158-f002:**
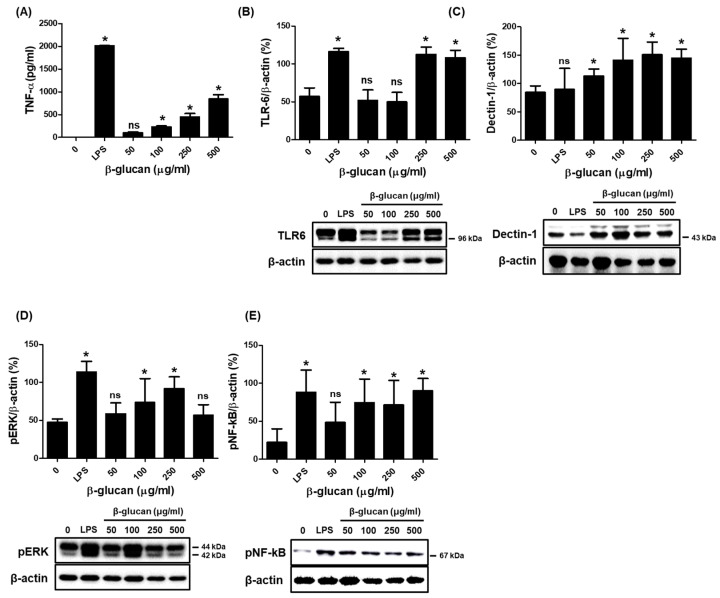
Effect of β-1,3-glucan on TNF-α production and the activation-related proteins in RAW264.7 cells. Cells were incubated with different concentrations (0, 50, 100, 250, and 500 μg/mL) of β-1,3-glucan or LPS for 24 h. The supernatants and cells were then harvested separately for each experiment. (**A**) TNF-α by ELISA assay and the expression of (**B**) TLR-6, (**C**) Dectin-1, (**D**) pERK, and (**E**) pNF-κB by Western blot analysis were identified. All data are presented as means ± SD, and experiments were performed at least three times. * *p* < 0.05 vs. untreated; ns: not significant.

**Figure 3 nutrients-16-03158-f003:**
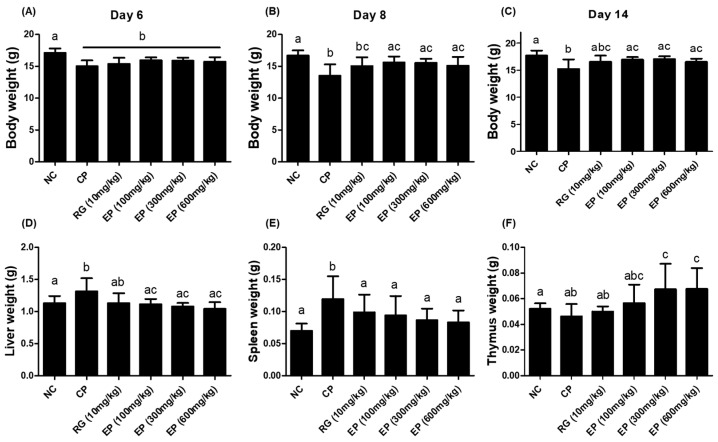
Change in body weight and liver, spleen, and thymus weights of the mice after oral administration of EP. Mice were injected intraperitoneally with CP on days 1 and 4 after starting oral administration of water, RG (10 mg/kg), and three concentrations (100, 300, and 600 mg/kg) of EP, except for the group receiving water only (NC). All specimens were administered orally to the mice for a total of 14 days. (**A**–**C**) Body weight of mice on days 6, 8, and 14 and the weight of (**D**) liver, (**E**) spleen, and (**F**) thymus isolated from each animal after 14 days of oral administration. All data are presented as means ± SD (n = 8) and analyzed by one-way ANOVA to compare differences between groups. The different letters in the graph indicate a significance between groups, and the *p*-value is less than 0.05.

**Figure 4 nutrients-16-03158-f004:**
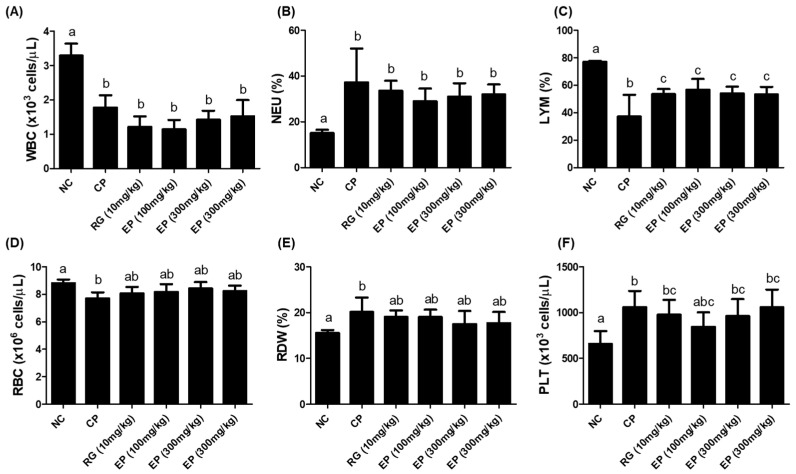
Effect of EP on peripheral blood cells of immunosuppressed mice. Whole blood was collected from mice after oral administration of each sample for 14 days. (**A**) White blood cells (WBC, ×10^6^ cells/μL), (**B**) neutrophils (NEU, %), (**C**) lymphocytes (LYM, %), (**D**) red blood cells (RBC, ×10^6^ cells/μL), (**E**) red blood cell distribution width (RDW, %), and (**F**) platelets (PLT, ×10^3^ cells/μL). All data are presented as means ± SD (n = 8) and analyzed with one-way ANOVA to compare differences between groups. The different letters in the graph indicate a significance between groups, and the *p*-value is less than 0.05.

**Figure 5 nutrients-16-03158-f005:**
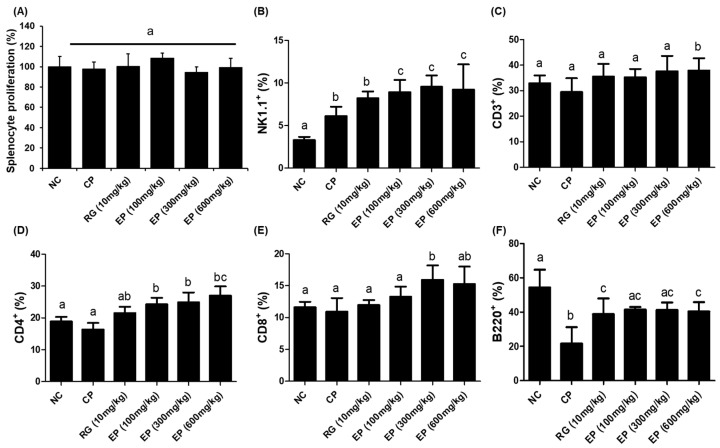
Effect of EP on splenocytes of immunosuppressed mice. (**A**) Splenocyte proliferation by CCK-8 assay and the frequencies of (**B**) NK1.1^+^, (**C**) CD3^+^, (**D**) CD4^+^, (**E**) CD8^+^, and (**F**) B220^+^ cells by flow cytometry were determined. All results are expressed as means ± SD (n = 8). Data were analyzed by one-way ANOVA to compare differences between groups. The different letters in the graph indicate a significance between groups, and the *p*-value is less than 0.05.

**Figure 6 nutrients-16-03158-f006:**
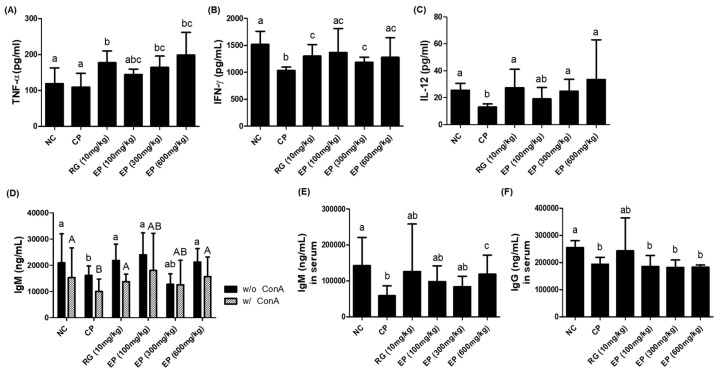
Effect of EP on the production of cytokines and antibodies. Splenocytes isolated from mice were incubated with ConA (10 μg/mL) for 72 h, and the cell-free supernatants were then collected, and mouse serum was separated from whole blood. The levels of cytokines and antibodies were analyzed by ELISA assay kits. The production of (**A**) TNF-α, (**B**) IFN-γ, (**C**) IL-12, and (**D**) IgM in the cell-free supernatants and the levels of (**E**) IgM and (**F**) IgG in the serum were determined. Data were analyzed by one-way ANOVA to compare differences between groups. The different letters in the graph indicate a significance between groups, and the *p*-value is less than 0.05.

**Figure 7 nutrients-16-03158-f007:**
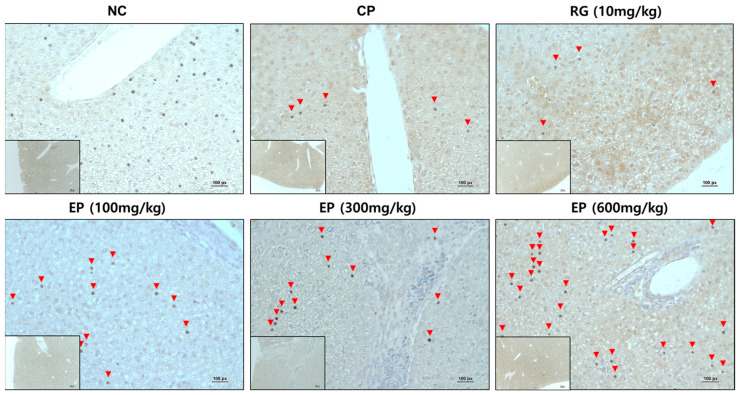
Effect of EP on dectin-1 expression in mouse liver. The liver tissue of mice was collected after 14 days of specimen treatment. The analysis of dectin-1 expression in the liver tissue of mice was performed by immunohistochemistry (IHC) staining. The picture was visualized by light microscopy (40× and 200×). Dectin-1 expression is indicated by black dots, shown by red arrows. The experiment was performed at least three times on different liver tissues from each group, and the upper picture was a representative of all results.

**Table 1 nutrients-16-03158-t001:** Effect of EP on hematological indices in CP-treated mice.

Hematological Index ^1^	Group NC	CP	RG (10 mg/kg)	EP (100 mg/kg)	EP (300 mg/kg)	EP (600 mg/kg)
WBC (×10^3^ cells/μL)	3.40 ± 0.68 ^a^	1.68 ± 0.30 ^b^	1.48 ± 0.31 ^b^	1.12 ± 0.30 ^b^	1.34 ± 0.27 ^b^	1.52 ± 0.47 ^b^
NEU (%)	14.95 ± 2.21 ^a^	41.40 ± 17.51 ^b^	37.29 ± 8.35 ^b^	31.99 ± 7.32 ^b^	33.10 ± 6.41 ^b^	36.53 ± 10.23 ^b^
LYM (%)	75.90 ± 2.18 ^a^	37.20 ± 15.88 ^b^	53.55 ± 3.64 ^c^	56.75 ± 7.83 ^c^	54.00 ± 4.96 ^c^	55.23 ± 5.54 ^c^
MONO (%)	5.97 ± 0.37 ^a^	11.17 ± 5.44 ^a^	10.08 ± 2.16 ^a^	10.36 ± 5.17 ^a^	7.84 ± 3.30 ^a^	7.96 ± 1.97 ^a^
EOS (%)	3.26 ± 1.65 ^a^	5.43 ± 5.23 ^a^	4.00 ± 3.16 ^a^	6.28 ± 2.63 ^a^	6.27 ± 4.77 ^a^	4.70 ± 3.54 ^a^
BASO (%)	0.10 ± 0.00 ^a^	0.20 ± 0.15 ^a^	0.14 ± 0.07 ^a^	0.11 ± 0.04 ^a^	0.13 ± 0.05 ^a^	0.14 ± 0.07 ^a^
RBC (×10^6^ cells/μL)	8.59 ± 0.49 ^a^	7.84 ± 0.44 ^b^	8.09 ± 0.48 ^ab^	8.17 ± 0.46 ^ab^	8.26 ± 0.46 ^ab^	8.27 ± 0.40 ^ab^
HGB (g/dL)	12.83 ± 0.80 ^a^	11.41 ± 0.85 ^a^	11.76 ± 0.94 ^a^	12.24 ± 0.77 ^a^	12.04 ± 1.01 ^a^	12.03 ± 0.63 ^a^
HCT (%)	40.78 ± 3.32 ^a^	36.39 ± 3.19 ^a^	38.36 ± 1.80 ^a^	39.15 ± 1.10 ^a^	38.83 ± 1.47 ^a^	38.56 ± 1.21 ^a^
MCV (fL)	47.46 ± 1.87 ^a^	46.44 ± 3.28 ^a^	47.55 ± 2.40 ^a^	48.05 ± 2.91 ^a^	47.08 ± 1.76 ^a^	46.78 ± 1.21 ^a^
MCH (pg)	14.96 ± 0.1 ^a^ 4	14.56 ± 0.78 ^a^	14.55 ± 0.42 ^a^	15.00 ± 0.33 ^a^	14.59 ± 0.48 ^a^	14.56 ± 0.51 ^a^
MCHC (g/dL)	31.56 ± 1.13 ^a^	31.43 ± 1.21 ^a^	30.64 ± 1.62 ^a^	31.29 ± 1.61 ^a^	31.04 ± 1.77 ^a^	31.18 ± 1.41 ^a^
RDW (%)	15.74 ± 0.63 ^a^	20.13 ± 3.14 ^b^	19.05 ± 1.43 ^ab^	19.03 ± 1.61 ^ab^	17.48 ± 2.87 ^ab^	17.72 ± 2.41 ^ab^
MPV (fL)	5.48 ± 0.31 ^a^	5.66 ± 0.59 ^a^	5.74 ± 0.57 ^a^	5.83 ± 0.51 ^a^	5.65 ± 0.44 ^a^	5.61 ± 0.49 ^a^
PLT (×10^3^ cells/μL)	734.5 ± 169.08 ^a^	1013.85 ± 177.21 ^b^	993.62 ± 161.95 ^bc^	845.12 ± 156.75 ^abc^	970.25 ± 185.14 ^bc^	1073.12 ± 194.64 ^bc^

^1^ WBC: white blood cells; NEU: neutrophils; LYM: lymphocytes; EOS: eosinophils; BASO: basophils; RBC: red blood cells; HGB: hemoglobin; HCT: hematocrit; MCV: mean corpuscular volume; MCH: mean corpuscular hemoglobin; MCHC: mean corpuscular hemoglobin concentration; RDW: red blood cell distribution width; MPV: mean platelet volume; PLT: platelet count. All data are means ± SD (n = 8) and were analyzed with one-way ANOVA to compare differences between groups. The different letters in the graph indicate a significance between groups, and the *p*-value is less than 0.05.

## Data Availability

All the data are contained within the article.

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
