# Peer review of "Euglena gracilis Enhances Innate and Adaptive Immunity through Specific Expression of Dectin-1 in CP-Induced Immunosuppressed Mice"

_nutrients, 2024, doi:10.3390/nu16183158_

Round 1
Reviewer 1 Report
Comments and Suggestions for Authors
The manuscript entitled "Euglena gracilis enhances innate and adaptive immunity through specific expression of dectin-1 in CP-induced immunosuppressed mice" by Hwan Hee Lee et al. presents a comprehensive study on the immune-enhancing effects of Euglena gracilis in cyclophosphamide (CP)-induced immunosuppressed mice. The study provides valuable insights into the role of β-1,3-glucan (paramylon) from E. gracilis in modulating both innate and adaptive immune responses. The findings are relevant and contribute to the understanding of natural immune stimulators. The experimental design is generally robust, and the results are clearly presented. However, there are a few areas where the manuscript could be improved.
1. The abstract is concise but could be improved by including a brief mention of the key results with some data, such as the specific cytokines affected by EP treatment, and the potential implications of these findings.
2. The introduction provides a good background but could be enriched with more information about the specific challenges of immunosuppression and how E. gracilis addresses these challenges.
3. The discussion section could be strengthened by including more recent studies that explore the role of β-glucans in immune modulation. This would help position the findings of this study within the broader context of current research on natural immune modulators.
Author Response
- In Abstract, the authors seem to use too many vague expressions that are not actual observations. For example, it is suggested to describe observations such as increase or decrease related or associated.
> Thank you for your advice, we modified Abstract as below
In line 25-27, “E.gracilis could be a good candidate for a natural immune stimulator in the innate and adaptive response by secreting TNF-α, IFN-γ and IL-12 through stimulating dectin-1 expression of the surface of immune cells.”
- The introduction provides a good background but could be enriched with more information about the specific challenges of immunosuppression and how E. gracilis addresses these challenges.
> Thanks for your comment, we added the explanation of why this study is conducted using CP-induced immunosuppressed mice as below
In line 71-77, “Several studies have demonstrated that the effect of E. gracilis on innate and adaptive immunity through in vitro and in vivo experiments, but the mechanism is still unclear. The immune system is critical to maintaining body homeostasis, so it has to be controlled to ensure that immune responses do not become excessively elevated or depressed. CP is known to induce the depression of lymphocytes in mice, which disrupts the homeostasis of the immune system. Considering that, we expected the immune-enhancing effect of EP containing large amounts of β-1,3-glucan paramylon in CP-induced immunosuppressed mice.”
- The discussion section could be strengthened by including more recent studies that explore the role of β-glucans in immune modulation. This would help position the findings of this study within the broader context of current research on natural immune modulators.
> Thanks for your opinion sincerely.
We have already mentioned in lines 357-383, the effect of β-glucan paramylon reported by several studies recently. The immunomodulating effects of β-glucan paramylon have been recognized only lately, and its conformation is different from other β-glucan derived from yeast and oats, resulting in different bioactivities. In addition, we would like to focus more on the immune-enhancing effect of E.gracilis containing a large amount of β-glucan paramylon in this study because of the challenge of developing as a bioproduct of E.gracilis for immune-enhancing effect.
Reviewer 2 Report
Comments and Suggestions for Authors
The present investigated the immune-enhancing effect of E. gracilis powder (EP) using RAW 264.7 cells and cyclophosphamide-induced immunosuppressed C57BL/6 mice. The author has done a lot of work and achieved certain results. But there are still the following issues need to be resolved.
1. Whether Dectin-1 have a regulatory relationship with NF-κB and ERK. What is the purpose of the author measuring the protein expressions of NF-κB and ERK at the cellular level? Please provide relevant descriptions.
2. In lines 139-143, What is the basis for constructing immunosuppressive mice? Why do mice need to receive cyclophosphamide again through drinking water after intraperitoneal injection? Can cyclophosphamide ingested orally reach the body in its original structure and exert immunosuppressive activity? What is the dosage of cyclophosphamide in drinking water?
3. What is the number of mice in each group?
4. The NC group in Figure 7, the enlarged image does not match the original image. According to figure 7C, it cannot be concluded that the dectin-1 expression in the CP group is lower than that in the NC group.
5. Why does the author detect Dectin-1 expression in liver tissue instead of important immune organs such as spleen and thymus. Is there any significant change in the liver? I did not see any relevant data in the article.
6. The pathological changes in the spleen, thymus, and organs are unknown. It is recommended that the author provide additional pathological images.
Author Response
- Whether Dectin-1 have a regulatory relationship with NF-κB and ERK. What is the purpose of the author measuring the protein expressions of NF-κB and ERK at the cellular level? Please provide relevant descriptions.
> Thanks for your comment.
Dectin-1 is one of pattern recognition receptor (PRRs) and ERK is one among its downstream molecules, and NF-kB is a transcription factor that control transcription of cytokine production. Therefore, we try to identify whether dectin-1 expression by β-glucan paramylon triggers its downstream molecules and transcription factor.
- In lines 139-143, What is the basis for constructing immunosuppressive mice? Why do mice need to receive cyclophosphamide again through drinking water after intraperitoneal injection? Can cyclophosphamide ingested orally reach the body in its original structure and exert immunosuppressive activity? What is the dosage of cyclophosphamide in drinking water?
> Thank you for your feedback and comments on our study.
The immune system shouldn't work in a normal state (if it does, it could be an autoimmune disease). Therefore, we thought it was correct to use the immunosuppressed mice to prove the immune-enhancing effect.
In this study, CP was only injected intraperitoneally twice on day 1 and 3, except for the control group (NC; oral administration of drinking water without CP). The repeated CP was designed to precisely suppress the immunity of the mice. The original role of CP in the body is a drug for cancer treatment, but its other working reduce the number of lymphocytes, so that inhibit immunity. This was described in line 64-70.
- What is the number of mice in each group?
> The number of mice in each group is eight (n=8). This was described in line 147.
- The NC group in Figure 7, the enlarged image does not match the original image. According to figure 7C, it cannot be concluded that the dectin-1 expression in the CP group is lower than that in the NC group.
> We're sorry for the misunderstanding.
We don't know why the results in Figure 7, the original image, don't match when you zoom in. We supported the original image for Figure 7.
In addition, we're sorry that you misunderstood on Figure 7 because we didn't mark the image of NC group with red arrows because there were many black dots. The NC group has more black dots representing dectin-1 expression than the CP group.
- Why does the author detect Dectin-1 expression in liver tissue instead of important immune organs such as spleen and thymus. Is there any significant change in the liver? I did not see any relevant data in the article.
> Thanks for your comments.
This study showed in vitro experiments, the specific expression of dectin-1 by β-glucan paramylon in mouse macrophage RAW264.7 cells (Figure 2C). The dectin-1 is expressed mainly on myeloid cells especially phagocytes (described in line 58-59), which present many in liver. In addition, it has been reported in previous study that dectin-1 is expressed in liver (described in ling 462-463). Therefore, we conducted to examine the dectin-1 expression in liver tissues.
- The pathological changes in the spleen, thymus, and organs are unknown. It is recommended that the author provide additional pathological images.
> Thanks for your opinion sincerely.
We will support the images of spleen and thymus of mice as Supplementary figures. We added the sentences “Figure S3: Changes of in the spleen and thymus of EP-treated mice” in line 481.
Reviewer 3 Report
Comments and Suggestions for Authors
The manuscript titled "Euglena gracilis enhances innate and adaptive immunity through specific expression of dectin-1 in CP-induced immunosuppressed mice" reports the immune-stimulating effect of Euglena gracilis. However, the authors should consider the following points: 1. Why does the title and body of the manuscript mention a study on Euglena gracilis when the experiments appear to have been conducted using a commercial β-1,3-glucan compound? Shouldn’t the title reflect the focus on the polysaccharide rather than the alga itself? 2. The results and discussion sections should be reviewed to ensure that immune effects are not attributed directly to Euglena gracilis if they were actually observed from the commercial β-1,3-glucan. 3. How did treatment with different concentrations of β-1,3-glucan affect nitric oxide (NO) production in RAW264.7 cells? 4. Which pro-inflammatory cytokine showed a significant increase in RAW264.7 cells treated with β-1,3-glucan? 5. Which cell surface proteins were affected by β-1,3-glucan treatment, and how did they influence cellular signaling? 6. How did treatment with Euglena gracilis (EP) affect the body weight of cyclophosphamide (CP)-induced immunosuppressed mice compared to the control group? Were there differences in the weight of the liver, spleen, and thymus? Why? 7. In the complete blood count (CBC), how did EP treatment influence neutrophil, lymphocyte, and other blood cell counts in CP-immunosuppressed mice? 8. What effect did EP treatment have on the frequency of NK1.1+, CD3+, CD4+, CD8+, and B220+ cells in the mice? 9. How did EP treatment affect the production of TNF-α, IFN-γ, and IL-12 in splenocytes from immunosuppressed mice? 10. What was the effect of EP treatment on IgM levels in the serum of immunosuppressed mice? 11. How did EP treatment influence the expression of dectin-1 in the liver of CP-immunosuppressed mice compared to RG treatment or the control group? 12. Based on the results obtained, how do the authors explain the immune-stimulatory effect of Euglena gracilis (EP) powder in immunosuppressed mice? 13. What molecular mechanisms are proposed for the activation of innate and adaptive immune responses in the presence of β-1,3-glucan? 14. Why do the authors believe that the production of TNF-α was significantly stimulated by β-1,3-glucan, while concentrations of IL-6 and IL-1β did not show a similar increase? 15. What is the relevance of the specific regulation of dectin-1 expression by EP in the liver of immunosuppressed mice, and how might this relate to the improvement of immune responses? 16. Based on the study results, why do the authors consider Euglena gracilis a good candidate as a natural immune stimulator for innate and adaptive responses? Since this alga contains a variety of bioactive compounds, don’t they think the conclusions should be more specific, given that only the effect of the β-1,3-glucan polysaccharide was evaluated and not the entire plant? How do they plan to address this limitation in their study? 17. Since the experiments were conducted with purchased β-1,3-glucan and not with direct extracts of Euglena gracilis, should the conclusions about the plant’s immune-stimulatory potential be revised or reformulated?
Comments on the Quality of English Language
The manuscript should be thoroughly reviewed for consistency in language use, as it currently contains parts written in both British and American English. Please standardize the language throughout the entire document.
Author Response
- Why does the title and body of the manuscript mention a study on Euglena gracilis when the experiments appear to have been conducted using a commercial β-1,3-glucan compound? Shouldn’t the title reflect the focus on the polysaccharide rather than the alga itself?
> Thanks for your opinion.
- gracilis can produce β-1,3-glucan in its cytoplasm through metabolic process (described in line 49-50) and E. gracilis contains a large amount of polysaccharide (β-1,3-glucan) as well as amino acid, pro-vitamin and lipids, therefore recent market tries to develop E. gracilis powder rather than a large β-1,3-glucan alone for inducing benefits to various physiological activity. Therefore, β-1,3-glucan is main effector in E.gracilis for immune-enhancing, but, we are considering to develop it as a bioproduct of E.gracilis powder itself rather than β-1,3-glucan.
In addition, this study included the results of in vitro and in vivo experiments. β-1,3-glucan was only used for in vitro experiments.for in vivo experiment, E. gracilis powder was orally given to mice. Therefore, we focused more on E. gracilis in this study.
- The results and discussion sections should be reviewed to ensure that immune effects are not attributed directly to Euglena gracilis if they were actually observed from the commercial β-1,3-glucan.
> In results and discussion, we described separately about the results of in vitro and in vivo experiments.
- gracilis was directly used for in vivo experiment and thus, we thought to be able to describe that E.gracilis contributed the immune-enhancing in immune system of mice.
- How did treatment with different concentrations of β-1,3-glucan affect nitric oxide (NO) production in RAW264.7 cells?
> As shown in Figure 1B, high concentrations (250 and 500 μg/ml) of β-1,3-glucan produced the NO in RAW264.7 cells. These results demonstrated that β-1,3-glucan induced the innate immune response via macrophage.
Which pro-inflammatory cytokine showed a significant increase in RAW264.7 cells treated with β-1,3-glucan?
> As shown in Figure 2A, the inflammatory cytokine TNF-α was significantly produced by RAW264.7 cells at the concentrations (100, 250 and 500 μg/ml) of β-1,3-glucan in RAW264.7 cells.
- Which cell surface proteins were affected by β-1,3-glucan treatment, and how did they influence cellular signaling?
> As shown in Figure 2B-C, β-1,3-glucan could bind TLR-6 and dectin-1 on the surface of mouse macrophage RAW264.7, which influenced the activation of its downstream molecule, ERK and the transcription factor, NF-kB.
- How did treatment with Euglena gracilis (EP) affect the body weight of cyclophosphamide (CP)-induced immunosuppressed mice compared to the control group? Were there differences in the weight of the liver, spleen, and thymus? Why?
> As shown in Figure 3A-C, CP injection reduced the weight of mice and E. gracilis (EP) rescued the effect of CP in mice.
CP injection increased the weight of liver and spleen. We think that the reason is that CP injection give toxic to mice. It has been known that enlarged spleen and liver could be caused by infection or liver diseases.
Whereas, CP injection slightly decreased the weight of thymus of mice. Thymus is an organ to develop T lymphocytes. These results could be connected with the results in Figure 5C-E that slightly decreased the frequencies of CD3+, CD4+ and CD8+.
- In the complete blood count (CBC), how did EP treatment influence neutrophil, lymphocyte, and other blood cell counts in CP-immunosuppressed mice?
> As shown in Figure 4, EP treatment did not affect WBC, neutrophil frequency, and others except for lymphocyte frequency. The frequency of lymphocytes was increased by EP treatment (Figure 4C).
- What effect did EP treatment have on the frequency of NK1.1+, CD3+, CD4+, CD8+, and B220+ cells in the mice?
> As shown in Figure 5B-F, EP treatment affects the increase in the population of NK1.1+, CD3+, CD4+, CD8+, and B220+.
- How did EP treatment affect the production of TNF-α, IFN-γ, and IL-12 in splenocytes from immunosuppressed mice?
> As shown in Figure 6A-C, EP treatment increased the production of TNF-α, IFN-γ, and IL-12 in splenocytes from immunosuppressed mice.
- What was the effect of EP treatment on IgM levels in the serum of immunosuppressed mice?
> As shown in Figure 6E, mice in the EP groups had increased serum IgM levels compared with mice in the CP group. This result showed that EP rescued the depression of B lymphocytes by CP injection. This result is related to the result of Figure 5F. In Figure 5F, the increased frequency of B220+ was shown by EP treatment.
- How did EP treatment influence the expression of dectin-1 in the liver of CP-immunosuppressed mice compared to RG treatment or the control group?
> Dectin-1 is mainly expressed on monocytes in the liver. Therefore, as shown in Figure 7, the expression of dectin-1 could be identified in liver tissue of mice of NC group.
However, the expression of dectin-1 was significantly decreased in CP group. RG treatment didn’t affect the expression of dectin-1 whereas, all concentration of EP treatment increased the expression of dectin-1 as much as the NC group.
- Based on the results obtained, how do the authors explain the immune-stimulatory effect of Euglena gracilis (EP) powder in immunosuppressed mice?
> As mentioned in the conclusion (lines 472-477), we consider that E. gracilis powder enhances the innate and adaptive immune response in immunosuppressed mice.
- What molecular mechanisms are proposed for the activation of innate and adaptive immune responses in the presence of β-1,3-glucan?
> In vitro experiment, β-1,3-glucan increased the activation of ERK and NF-kB in macrophages. Based on this result, we expect that β-1,3-glucan can activate the molecules associated with PRRs in innate and adaptive immune cells.
- Why do the authors believe that the production of TNF-α was significantly stimulated by β-1,3-glucan, while concentrations of IL-6 and IL-1β did not show a similar increase?
> We examined the production of TNF-α as well as IL-6 and IL-1β by macrophage RAW264.7 cells by treatment with β-1,3-glucan. However, IL-6 and IL-1β were not produced by macrophage RAW264.7 cells and we've provided the results as Supplementary Figure 1.
- What is the relevance of the specific regulation of dectin-1 expression by EP in the liver of immunosuppressed mice, and how might this relate to the improvement of immune responses?
> Dectin-1 is a type II transmembrane protein of the pattern recognition receptor expressed on myeloid such as dendritic cells, macrophages, neutrophils and monocytes. Leukocytes is developed in bone marrow and transferred to various tissues through blood. There are many macrophages (Kupffer cells) in liver tissue, and resident macrophages recognize ligands that come into the liver and work the defense mechanism against foreign. Therefore, we expect that the specific regulation of dectin-1 expression by EP is involved in the activation of innate and adaptive immune responses.
- Based on the study results, why do the authors consider Euglena gracilis a good candidate as a natural immune stimulator for innate and adaptive responses? Since this alga contains a variety of bioactive compounds, don’t they think the conclusions should be more specific, given that only the effect of the β-1,3-glucan polysaccharide was evaluated and not the entire plant? How do they plan to address this limitation in their study?
> As mentioned in the Introduction and Discussion, E. gracilis contains various components, amino acids, provitamins, lipids and polysaccharide (β-1,3-glucan), this was described in lines 51-53. However, the main source of E. gracilis is known to be β-1,3-glucan (> 55%), which is accumulated by metabolism.
In this study, E. gracilis was used it over 55% of β-1,3-glucan and as your comment, our sample will contain β-1,3-glucan as well as other components; however, we think that β-1,3-glucan in E. gracilis is major effector in immune-enhancing because of the dectin-1 expression on liver tissues of mice (as shown in Figure 7). β-1,3-glucan is a specific ligand to be recognized by dectin-1 on the macrophages that generate immune responses. This study also identified the expression of dectin-1 by β-1,3-glucan on macrophage (Figure 1B).
The reason suggested that E.gracilis is a good candidate of natural immune stimulator is because E.gracilis is found in natural, and our results could be evidence shown to immune-enhancing effect.
- Since the experiments were conducted with purchased β-1,3-glucan and not with direct extracts of Euglena gracilis, should the conclusions about the plant’s immune-stimulatory potential be revised or reformulated?
> Thanks for your advice sincerely.
For an in vitro experiment, we used the β-1,3-glucan purchased from Sigma-aldrich; however, the product is derived from E. gracilis and contains only β-1,3-glucan. For an in vivo experiment, we used E.gracilis, which contains over 55% of β-1,3-glucan.
Therefore, we cautiously suggest that this conclusion is correct for this study.
Round 2
Reviewer 2 Report
Comments and Suggestions for Authors
ok
Reviewer 3 Report
Comments and Suggestions for Authors
The authors of the manuscript 'Euglena gracilis enhances innate and adaptive immunity through specific expression of dectin-1 in CP-induced immunosuppressed mice' have responded and made the changes suggested by the reviewers. The manuscript has improved in quality and can be published in its current form.